

# Estimating plant biomass in agroecosystems using a drop-plate meter

Stephen M. Robertson, Ryan B. Schmid and Jonathan G. Lundgren

Ecdysis Foundation, Estelline, South Dakota, United States of America

## ABSTRACT

**Reason for doing the work:** Plant biomass is a commonly used metric to assess agricultural health and productivity. Removing plant material is the most accurate method to estimate plant biomass, but this approach is time consuming, labor intensive, and destructive. Previous attempts to use indirect methods to estimate plant biomass have been limited in breadth and/or have added complexity in data collection and/or modeling. A cost-effective, quick, accurate, and easy to use and understand approach is desirable for use by scientists and growers.

**Objectives:** An indirect method for estimating plant biomass using a drop-plate meter was explored for use in broad array of crop systems.

**Methods:** Drop-plate data collected by more than 20 individuals from 16 crop types on 312 farms across 15 states were used to generate models to estimate plant biomass among and within crop types.

**Results:** A linear model using data from all crop types explained approximately 67% of the variation in plant biomass overall. This model performed differently among crop types and stand heights, which was owed to differences among sample sizes and farming between annual and perennial systems. Comparatively, the model using the combined dataset explained more variance in biomass than models generated with commodity specific data, with the exception of wheat.

**Conclusions:** The drop-plate approach described here was inexpensive, quick, simple, and easy to interpret, and the model generated was robust to error and accurate across multiple crop types. The methods met all expectations for a broad-use approach to estimating plant biomass and are recommended for use across all agroecosystems included in this study. While it may be useful in crops beyond those included, validation is suggested before application.

## INTRODUCTION

Plant biomass is a reliable metric used to assess the health and productivity of agroecosystems. Plants convert carbon dioxide into sugars to generate new tissue, or biomass, sequestering carbon in plant material. In this way, plant growth is a potential pathway to decrease atmospheric carbon and mitigate the effects of climate change (*Dusenge, Duarte & Way, 2019*; *Terrer et al., 2019*). Plant biomass directly increases arthropod abundance and richness (*Wimp et al., 2010*; *Lu et al., 2021*), and when

Corresponding author
Jonathan G. Lundgren,
jonathan.lundgren@ecdysis.bio

appropriately managed, provides a number of benefits for agriculture, such as soil erosion control, improvements to soil chemical and physical properties, supporting diverse microbiomes, and improving production (*Sakar et al., 2020*). Importantly, plant biomass is the primary food source for livestock in pasture and rangeland systems. Measurements of plant biomass are used by growers to estimate available forage for livestock, allowing growers to make decisions on when and where to move animals (*Barnhart, 1998*; *Rayburn & Lozier, 2016*). Given the value of plant biomass to both scientists and growers, it is important to consider the efficiency and cost of methods that assess plant biomass such that changes can be easily monitored.

The most accurate and direct method to measure plant biomass involves taking clippings of above-ground plant material at multiple locations within an area, drying these samples over days to weeks, and weighing the dried plant material. This method is destructive, requiring plant material to be removed, and a major limitation for this approach is time and labor, both being relatively high (*Harmoney et al., 1997*). Multiple studies have established less-intensive methods to estimate plant biomass, including falling- and rising-plate meters (*Bransby, Matches & Krause, 1977*; *Rayburn & Rayburn, 1998*; *Rayburn & Lozier, 2016*; *Hart et al., 2020*), Robel poles (*Robel et al., 1970*), and canopy height sticks (*Michalk & Herbert, 1977*), with varying levels of success. More recently, researchers have attempted to use modern technologies, such as spectral data from hand-held devices (*Zhang et al., 2021*), aerial drones (*Lussem et al., 2019*; *Amorim et al., 2022*), and satellites (*Sankaran, Quirós & Miklas, 2019*; *Zumo, Hashim & Hassan, 2022*), alone or in combination with simpler measures to generate more accurate estimates of plant biomass (*Gargiulo et al., 2020*). While approaches that incorporate complex models or modern technology have advantages, they also reduce model flexibility and producer accessibility due to the added cost of equipment, complexity of data, and difficulty in data retrieval.

Drop-plate meters may offer a level of accuracy needed for research purposes, while also providing a simple and cost-effective tool that growers can use to estimate plant biomass for making decisions and monitoring improvements. This method involves placing a plate of given mass and area onto the plant canopy and measuring the displacement height of slightly compressed plant material within the plate area (*Bransby, Matches & Krause, 1977*; *Rayburn & Rayburn, 1998*). The entire apparatus can be constructed with minimal cost and has explained as much as 94% of the variability in plant biomass in grassland systems (*Zambatis et al., 2006*). This approach requires calibration and is typically employed in situations with predefined (*Hart et al., 2020*) or simple community compositions (*Griggs & Stringer, 1988*). Nonetheless, such an approach could be calibrated for use in a wide variety of systems with varying degrees of community complexity.

In a larger effort to detail the efficacy of regenerative agriculture (*sensu Lacanne & Lundgren, 2018*), an attempt was made to use a drop-plate meter to estimate plant biomass across multiple regions and farming systems, including row crops, specialty crops, and pastures. The goal was to calibrate this method for use among the studied systems and regions to achieve a large, accurate dataset using a user-friendly technique. To do so, we tested two primary hypotheses: Drop-plate meters can be used to accurately estimate plant

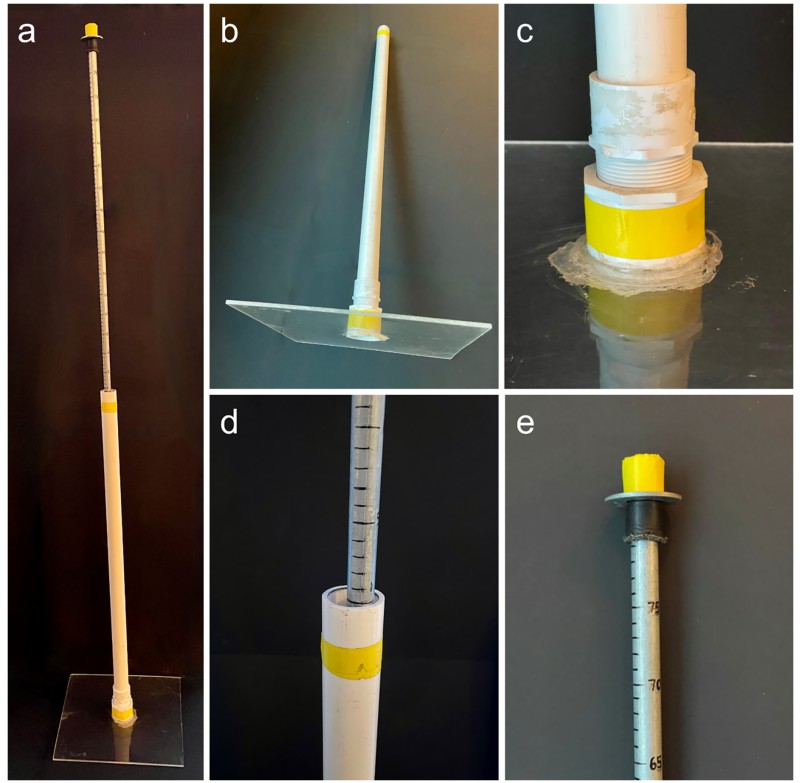

**Figure 1 Photos of the drop-plate meter.** (A) The fully assembled drop-plate meter. (B) The weighted portion (1.44 kg) of the drop-plate meter with 0.12 cm$^2$ plexiglass base. (C) The PVC drain super glued to the base plate with adapter attached. (D) The point where the metal rod meets the plate apparatus, *i.e.*, a reading of 0 cm. (E) The top of the metal rod showing the maximum reading of 80 cm.

biomass (1) among similar commodities and (2) across all agricultural systems. Here we detail the methods employed and accuracy obtained.

## METHODS

Plant biomass was measured on 312 farms across 15 states (CA, CO, IA, IL, IN, KS, MI, MN, MT, ND, NE, OR, SD, WA, and WI) in the United States (Fig. S1). Crop systems varied and included alfalfa, almonds, apples, barley, canola, cherries, corn, cover, grapes, kernza, livestock (tame-grass pastures), oats, soy, and wheat. Each farm was visited once during the growing season (spring for specialty crops and summer for livestock and row crop systems). We regard a pasture as a grassland used to produce livestock.

Drop plates were constructed using acrylic plexiglass, PVC, and a metal measuring rod (Fig. 1). The plate itself was made using 0.64 cm thick acrylic plexiglass cut into a 35.5 cm square. The plate was glued between a PVC floor drain (center removed to allow measuring rod through) and to a female threaded PVC adapter. This was screwed into a male threaded adapter attached to a 91 cm long, 3.18 cm diameter PVC (Type 1 SCH 40) pipe. The total height and weight of this apparatus was 98 cm and 1.44 kg, respectively. Into the center PVC pipe of the plate apparatus was put a metal measuring rod, which was

marked with permanent marker at every centimeter starting at the spot where the top of the PVC pipe met the inserted metal rod on a flat surface.

Plant biomass was recorded from four points (at least 15 m apart) in each field ($n$ = 1,248) that were at least 10 m from the field margin. Drop-plate measurements were taken by first resting the metal rod onto the soil surface. The plate apparatus was then placed onto the plant canopy and slowly allowed to come to rest, recording resting plate heights to the nearest half centimeter. Each plate reading was followed by collecting plant material beneath the plate. A PVC square (35.5 cm × 35.5 cm) was placed over the drop-plate apparatus to outline the area from which the drop-plate measurement was taken. All plant material from within the PVC square was cut at the soil interface and placed into a labeled paper bag. Samples were allowed to air dry for no less than 2 wks before weighing the dried plant material. Plant biomass weights were paired with drop-plate readings for each sample point.

Data were analyzed overall and within each commodity with an $n$ > 20 to generate and compare model effectiveness. Seasonal commodities of similar genera, such as spring and winter wheat (genus *Triticum*), were categorized as the same commodity in all analyses as these are assumed to have similar relationships between compression (drop-plate readings) and biomass. Each data set was treated independently. Incomplete data points (*i.e.*, those without either drop-plate or biomass measures) were removed from each data set. Data were also removed when one measure was zero and the other >0, as these data represented terrain (*i.e.*, drop plate > 0; biomass = 0) or previous crop residues (*i.e.*, drop plate = 0; biomass > 0). Outliers were first identified by generating boxplots for each drop-plate measure. Those data points that were beyond the measured variance within drop-plate heights were removed. Linear models were then generated for each grouping. Standardized residual errors (SREs) were calculated for each data point in each model. Those data points whose SRE were beyond an absolute value of three were removed as outliers. In the full model, absolute values of the remaining SREs were analyzed among commodity and drop-plate readings (drop-plate readings grouped 0–5, 5–10, 10–20, 20–40, and 40+ cm) using a Bonferroni corrected alpha value ($\alpha$ = 0.025). Because sample sizes from these groupings were very different, we used Kruskal–Wallis tests to determine if the model performed similarly across all commodities and drop-plated measures. Wilcoxon pairwise comparisons were used to determine which commodities and drop-plate categories performed best and worst. All analyses were performed in RStudio v2022.02.0 (*RStudio Team, 2022*).

## RESULTS

The drop-plate meter used in this study was constructed using materials easily obtained from a local hardware store for less than 20 USD. Use of the drop-plate meter was simple, easily trained and understood, and measurements took less than 5 s to acquire. Data were removed when crop residue was present and collected and when drop-plate measures included previous crop residues (primarily corn stalks). Of the original 1,248 data points, 919 (73.6%) data points from 16 known and one unknown commodity types were used in the overall model (see model summaries in Table 1). The final model described

**Table 1 Statistics for linear models generated from commodity data.**

| Commodity/System | $n$ | $\beta_0$ | $\beta_1$ | $r^2$ | RSE |
|---|---|---|---|---|---|
| All commodities | 919 | 12.87 | 2.50** | 0.669 | 19.52 |
| Almonds | 86 | 14.60 | 4.21** | 0.444 | 14.59 |
| Apples | 62 | 5.38 | 2.25** | 0.364 | 6.43 |
| Cherries | 61 | 1.48 | 5.49** | 0.506 | 13.35 |
| Corn | 89 | 18.64 | 1.95* | 0.117 | 26.96 |
| Grapes | 51 | 10.47 | 3.43** | 0.575 | 11.25 |
| Pastures | 291 | 9.58 | 3.15** | 0.550 | 17.62 |
| Soy | 58 | 54.49 | 0.74 | 0.032 | 46.6 |
| Wheat | 151 | 12.10 | 2.48** | 0.699 | 28.79 |

**Note:**

Provided are the sample size ($n$), y-intercept ($\beta_0$), slope ($\beta_1$), coefficient of determination ($r^2$), and residual standard error (RSE).

\* Significance of the slope at $\alpha = 0.05$.

\*\* $p < 0.0001$.

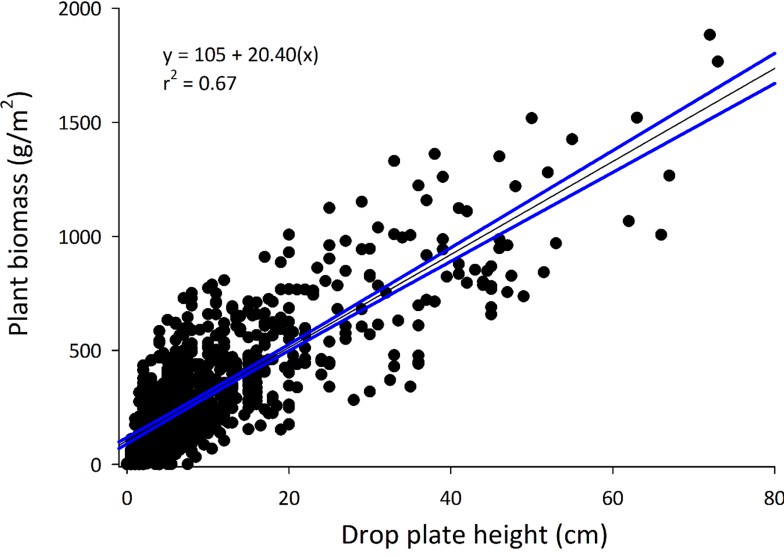

**Figure 2 The full model developed using all commodities.** The equation and $r^2$ value are provided on the graph. Note that the y-axis has been converted to $g/m^2$, producing a different y-intercept and slope than those provided in Table 1 (same coefficient of determination).

approximately 67% ($r^2 = 0.668$) of the variance in biomass measures across all commodities (Fig. 2). The generated line did not perform similarly among commodities ($\chi^2_{16} = 89.61$; $p = 2.96 \times 10^{-12}$; Table 2). The model described most commodities similarly well, with all except barley, oats, soy, and wheat relative standard errors (RSEs) being similar to that of grapes, which maintained the lowest mean RSE absolute values. Likewise, RSEs from the overall model differed between groupings of drop-plate readings ($\chi^2_4 = 76.22$; $p = 1.10 \times 10^{-15}$). In general, as the drop-plate readings increased, the model performed decreasingly well, with the 0–5 cm being most well described and 20–40 and 40+ cm being least well described.

**Table 2 Comparisons of mean absolute standard residual errors among commodities and drop-plate groupings.**

| Commodity | n | Mean |SRE| |
|---|---|---|
| Grapes | 52 | 0.427 (A) |
| Apples | 72 | 0.478 (AB) |
| Cherries | 67 | 0.600 (AB) |
| Pasture | 298 | 0.669 (AB) |
| Almonds | 92 | 0.587 (AB) |
| Unknown | 4 | 0.301 (ABC) |
| Lentils | 4 | 0.668 (ABC) |
| Canola | 2 | 1.437 (ABC) |
| Flax | 4 | 0.978 (ABC) |
| Cover | 10 | 0.748 (ABC) |
| Alfalfa | 9 | 1.042 (ABC) |
| Kernza | 14 | 0.764 (ABC) |
| Corn | 72 | 0.686 (ABC) |
| Soy | 34 | 0.940 (ABC) |
| Barley | 16 | 0.772 (BC) |
| Oats | 16 | 1.389 (C) |
| Wheat | 153 | 0.877 (C) |
| **Drop-plate group** | **N** | **Mean |SRE|** |
| 0–5 cm | 422 | 0.554 (A) |
| 5–10 cm | 232 | 0.671 (AB) |
| 10–20 cm | 161 | 0.800 (B) |
| 20–40 cm | 71 | 1.126 (C) |
| 40+ cm | 33 | 1.223 (C) |

**Note:**
Relationships are provided in parentheses after means.

Models generated for each commodity with $n > 20$ varied in performance (see Table 1). Only the model generated for soy was not significantly positive ($p = 0.178$). Notably, the model created using data from wheat ($n = 151$) described the greatest amount of variance in plant biomass collected in wheat fields ($r^2 = 0.699$), and the models created using corn ($n = 89$) and soy ($n = 58$) data described the least amount of variance in biomass collected in each field type ($r^2_{corn} = 0.117$; $r^2_{soy} = 0.032$). Standardized residual errors for models generated for corn and soy were 26.96 and 46.67, respectively.

## DISCUSSION

The drop-plate meter and data collected using it met all expectations, being cost effective, easy to construct and operate, quick to deploy, and proving to generate reliable and robust predictors of plant biomass. This approach's effectiveness is evidenced by its continued use in other habitats around the world (*Vermeire et al., 2021*; *Parker et al., 2023*). The model generated using data from all crop systems showed that drop-plate measurements significantly predicted plant biomass across a wide swath of agroecosystems and

geographic regions that varied substantially in plant community characteristics. The resulting relationship was robust enough to accommodate plant stand complexity, varying terrain, and multiple users. Previous linear models generated to predict plant biomass using drop-plate (or rising-plate) heights have achieved $r^2$ values of 0.79–0.94 (*Bransby, Matches & Krause, 1977*), 0.09–0.84 (*Vartha & Matches, 1977*), 0.90–0.94 (*Griggs & Stringer, 1988*), 0.74 (*Trollope & Potgieter, 1986*), 0.52 (*Rayburn & Rayburn, 1998*), 0.61 (*Zambatis et al., 2006*), and 0.76 (*Harmes, Dreber & Trollope, 2019*). These model performances were developed using data with stricter criteria and/or more focused research systems, such as predicting biomass within seasonal growth periods, specific drop-plate heights, or single commodities (*e.g.*, only alfalfa or tall fescue), than the model presented here. *Griggs & Stringer (1988)* added additional independent variables, such as ground cover and days of growth, to their models to improve model performance. *Zambatis et al. (2006)* fitted multiple non-linear models, such as Hoerl, Gaussian, and sinusoidal, to improve model performance. While higher $r^2$ values were often achieved using these approaches, the simplicity in understanding and broad applicability are diminished with each subset category and as modelling complexity increases, and the sources for error are increased as independent variables are added. The model presented here is designed to be simple to understand and apply, broadly applicable, and having an accuracy that falls in the upper-middle range of performances among previous modeling attempts.

The model including all crops was applied to individual commodities with mixed results. Commodities with the lowest sample sizes ($\leq$10) were similar to all other commodities with respect to RSEs. This likely resulted from a lack of resolution in the variation between drop-plate measures and biomass. Commodities with sample sizes between 11 and 20 (barley, kernza, and oats) contributed relatively little to the overall model, and as such, RSEs were consistently high. Interestingly, drop-plate measures were less related to plant biomass in row crops and pastures than perennial fruits. Row crops generally produced higher drop-plate measures and plant biomass than in orchards (this test not reported due to the relationship with drop-plate groupings); apples had the only drop-plate reading above 20 cm (23 cm) among all fruits. Plant communities were relatively short ($\leq$10 cm drop-plate readings) in most studied crop systems (654/919 or 71.2% of samples) and thus these shorter plant communities had a strong contribution to the overall model. As such, these measures had the lowest RSEs among drop-plate groupings. Higher drop-plate values (>20 cm) were relatively rare (104/919 or 11.3% of samples). This resulted in much lower definition of the variance at these extreme measures and model accuracy degraded at drop-plate measures >20. Taller plant communities have showed higher variability in biomass in other work (*Bransby, Matches & Krause, 1977*; *Griggs & Stringer, 1988*), suggesting greater complexity in communities with taller plants. While including more drop-plate heights above 20 cm would likely improve model accuracy at this upper extreme, these measures are not common enough to be represented in numbers comparable to lower measures. In fact, *Zambatis et al. (2006)* reported mean and median drop-plate heights of model outliers as 35.4 and 32.8 cm, respectively, by comparison to 18.7 and 16.5 cm for non-outliers, and the models generated by these

researchers tended to decrease in effectiveness at these higher ranges regardless of model complexity. Nonetheless, the model generated here using all data tracked reasonably well with extremely high drop-plate measures.

Models generated using commodity-specific data tended to describe less variance in plant biomass than the full model, with the exception of wheat. Crop spacing varied considerably among operations, which added variation and forced researchers to measure crops within drop-plate areas. In each commodity, model optimization suffered from poor definition in biomass variation among drop-plate measures. As such, data outliers were not well recognized. Plant biomass in corn and soy was particularly poorly related to drop-plate readings. These crops were often grown in rotation, with corn stalks and crop residues commonly left in fields. Additionally, these systems are often heavily tilled, with turned soil adding to drop-plate errors. Although outliers caused by these errors were identified well in the full model, commodity-specific models did not provide the sample size to recognize extreme data points as outliers. This is evident in differences in commodity loadings for the full model and commodity-specific models. For example, corn had 72 and soy had 34 data points included in the full model, while 89 and 58 corn and soy data points were included in the commodity-specific models. The result was higher than average RSEs and lower than average $r^2$ values among commodity specific models.
In contrast, the model generated specifically for wheat achieved the highest coefficient of variation among all generated models. Wheat was grown in a relatively consistent manner, in terms of spacing and density, and crop residues were not present in studied fields, which produced highly consistent data. Most data points (163 total) from wheat were kept in modeling, both in the full ($n = 153$) and commodity-specific model ($n = 151$), highlighting the consistency in the data. Although more descriptive and accurate to wheat, this model is restrictive in application and the full model recommended for use in wheat due to its simplicity.

## CONCLUSION

The described approach to estimating biomass is recommended for broad use in agricultural systems. Data collection and the model generated using combined commodity data met all desired criteria. The drop-plate meter was inexpensive, easy to build and use, and data using the apparatus was collected rapidly. The resulting overall model was accurate across many commodities, robust to error associated with commodities and users, and simple to understand. Measurements were collected once during the growing season from farms whose crops were in a variety of growth stages, providing model flexibility that allows growers and researchers to employ this approach at any time. Given the speed at which these methods can be employed, data can be collected and decisions made in real time, removing the need to forecast biomass levels based on plant-growth averages that may or may not apply to or may or may not be available in specific circumstances. Overall, this is a powerful method to indirectly estimate plant biomass in agroecosystems.

Important caveats exist with the use of this methodology. The weight of drop-plates is important to the efficacy of this approach and should be replicated as closely as possible (drilling small holes in the drop-plate as needed is acceptable). The generated model is

intended only to estimate vertical biomass (including senesced plant material) and will not accurately estimate biomass of existing residues. Drop-plate readings that are affected by terrain and crop residues are not appropriate for estimating plant biomass with this approach and should be avoided. The drop-plate described here covers approximately $1/8^{th}$ of a square meter ($0.123$ m$^2$) and conversions are necessary to extrapolate to larger areas. Plant heights, and thus drop-plate readings, are not uniform across a field. Drop-plate measurements should be collected from multiple locations across an area to obtain a representative average of the area under investigation. It is suggested to take no fewer than two readings from random locations per 4,000 m$^2$ (~one acre), or three measures per hectare, to obtain an average that accounts for variation within fields. More measures are recommended for highly variable stand heights or when more accurate estimations of biomass within fields or paddocks are desired. The use of this approach in commodities not represented in this study, while likely to perform well, should be validated prior to application.

## ACKNOWLEDGEMENTS

Three hundred and fifteen farmers were registered volunteers for this study. We would like express gratitude to the staff at Ecdysis Foundation, specifically N. Bell, M. Bredeson, K. Busenitz, R. Butterbaugh, K. Clinton, T. Fenster, A. Heibult, W. Hillery, M. Jones, A. Knofczynski, C. Lind, A. Michels, C. Millar, D. Pecenka, A. Shorter, J. Skaar, A. Tess, O. Torbert, K. Welch, M. Werger, and A. Zhang for their help and hard work in data acquisition.

### Funding

This work was supported by #NoRegrets Initiative, Rockefeller Foundation, and General Mills. The funders had no role in study design, data collection and analysis, decision to publish, or preparation of the manuscript.

### Grant Disclosures

The following grant information was disclosed by the authors:
#NoRegrets Initiative.
Rockefeller Foundation.
General Mills.

### Competing Interests

All authors are employed by the Ecdysis Foundation.

### Author Contributions

- Stephen M. Robertson performed the experiments, analyzed the data, prepared figures and/or tables, authored or reviewed drafts of the article, and approved the final draft.
- Ryan B. Schmid performed the experiments, authored or reviewed drafts of the article, and approved the final draft.

- Jonathan G. Lundgren conceived and designed the experiments, performed the experiments, prepared figures and/or tables, authored or reviewed drafts of the article, and approved the final draft.

## Data Availability
The raw data is available in the Supplemental Files.

## Supplemental Information
Supplemental information for this article can be found online at http://dx.doi.org/10.7717/peerj.15740#supplemental-information.

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
