# Peer review of "Estimating plant biomass in agroecosystems using a drop-plate meter"

_PeerJ, doi:10.7717/peerj.15740_

## Round 0.1 · original submission · Major Revisions

Dear authors,

This manuscript is an interesting and potentially important topic, and the research presentation standard is good. However, many notes should be considered before accepting this manuscript as follows:

- The title should represent the article's content and facilitate retrieval in indices developed by secondary literature services. A good title (i) briefly identifies the subject, (ii) indicates the purpose of the study, and (iii) gives important and high-impact words early.

- The abstract must be completely self-explanatory and intelligible in itself. It should include the following: 1. Reason for doing work, including rationale or justification for the research; 2. Objectives and topics covered; 3. Brief description of methods used. If the paper deals mainly with methods, give the basic principles, range, and degree of accuracy for new methods; 4. Results; 5. Conclusions.

- I wonder how the author used the term “Commodity”. It should be a plant species. In addition, Pasture should be defined exactly. What types of plant species are included?

- Fig.2. It is needed to test for the polynomial model and to see the value of R2. Is it higher or lower than the R2 of the linear model?

Please pay particular attention to the detailed comments from Reviewer 1.

- Discussion: Discussion needs more effort with providing recent references.

Reviewer 1 ·

Basic reporting

The paper attempts to use a square plexiglass plate to standardize the measurement of standing crop biomass commodity crop plantings. The reporting of the purpose is clear, professionally structured and understandable.

What is missing from the article is a review of the standing crop biomass measurement literature. The literature is replete with alternative methods such as proposed here for standing crop biomass estimation. Because of this lack of foundation, the small size (35 cm x 35 cm) and shape (square) of the plot (plate) have largely been discounted within the literature to provide inconsistent measurements of standing crop biomass. This method also fails to accounted for rooted vs overlapping biomass which has been documented to contribute significant error, especially when comparing geometric and non-geometric crop growth patterns.

Experimental design

What is missing from the article is a review of the standing crop biomass measurement literature. The literature is replete with alternative methods such as proposed here for standing crop biomass estimation. Because of this lack of foundation, the small size (35 cm x 35 cm) and shape (square) of the plot (plate) have largely been discounted within the literature to provide inconsistent measurements of standing crop biomass. This method also fails to accounted for rooted vs overlapping biomass which has been documented to contribute significant error, especially when comparing geometric and non-geometric crop growth patterns.

It is common for example, in circular 1-2 meter square standing crop biomass samples in cropped and natural areas to achieve r2 values that routinely exceed 95%.

The small sample size (3 per field if I understand it correctly) coupled with the small plot size and shape of the plot would not provide reliable estimates of biomass on all crops at all stages of a crop life cycle. Because they have not standardized for crop sampling time within the life cycle of the crops, this adds considerable additional variability within a crop, accross fields of even the same crop. The temporal contribution to estimating crop type biomass does not appear to be evaluated in this paper.

Validity of the findings

Depending on the questions being asked, this method may be suitable. But, for robust biomass measurements that are evaluating energetics, nutritional mass balances of nutrients in plant biomass, or following the dynamics of crops and foraging/grazing wildlife (perhaps including insect herbivores) if this method has the high variances ( 30-70% R-squares) we would not find this useful for use in crop or non-cropland above ground standing crop biomass measurements.

Some additional thoughts on why this method would not be useful:

1. Results can not be scaled. For example, to be able to use this data to crosswalk the calibration of remote sensed landscape-scale (even farm scale) standing crop biomass is not possible. The small scale of the plots do not provide a sufficient target for remote sensing calibration.

2. Concern about the stated time and labor for sampling a herbaceous biomass plot of 1 meter square plot can be clipped and bagged in < 2 minutes even with physiogamy sampling using portable battery powered hedge trimmer. A 2 x 3 m plot requires on average 2-3 minutes. Drying requires 2 days maximum to 7-8% moisture.

Clipping woody vegetation crop plots (e.g. shrub) can take 4-5 minutes each for 1 sm plots.

Additional comments

Standard measurements can be streamlined and can greatly reduce time, labor, and significinatly improve estimation value and usefulness in addressing many types of research questions.

·

Basic reporting

This manuscript presented a good method for plant biomass that help researchers and growers to estimate crop or plant biomass

Experimental design

Good experimental design

Validity of the findings

The findings in this manuscript will help scientists and researcher for calculating plant biomass

Additional comments

The references in text can be write in another form i.e more than two researchers the 1st name write followed by et al.
The discussion need to newest references while, the newest one one 2006
also English check
The methods part needs to subtitles

·

Basic reporting

The article entitled (Estimating plant biomass across multiple agroecosystems and regions using a drop-plate meter) is interesting for plant workers. Getting an important clue to waste the time of conventional methods used for biomass estimation.
The manuscript is well written. It should be suitable for readers of Peerj and can be published as it is.

Experimental design

No comment

Validity of the findings

No comment

Additional comments

No comment

---

## Round 0.2 · accepted · Accept

Dear authors,

Thank you for responding to the comments raised by reviewers.

Best regards

Magdi Abdelhamid